# HSPS-10—Short Version of the Highly Sensitive Person Scale for Students Aged 12–25 Years

**DOI:** 10.3390/ijerph192315775

**Published:** 2022-11-27

**Authors:** Monika Baryła-Matejczuk, Robert Porzak, Wiesław Poleszak

**Affiliations:** Institute of Psychology and Human Sciences, WSEI University, 20-209 Lublin, Poland

**Keywords:** sensory processing sensitivity, highly sensitive person scale, environmental sensitivity, psychometric properties, normalized standard scores

## Abstract

The aim of the article is to present a short version of the Highly Sensitive Person Scale (HSPS-10) as a useful tool for the assessment of adolescents and young adults and to improve their self-awareness. (1) Background: The original American HSPS was developed as a tool for the assessment of Sensory Processing Sensitivity (SPS), which is understood to be an inherited temperamental trait. The basis for the research is the concept of SPS, which may be included within the broader construct of the Environmental Sensitivity (ES) model. (2) Methods: The research used a Polish-language, short version developed on the basis of the Highly Sensitive Person Scale, where the respondents answered 10 questions in a 7-point Likert scale. (3) Results: The test results show that the Polish, HSPS-10 is a reliable and valid measurement of the SPS construct and that the results obtained using the abbreviated version indicate a three-factor structure. The structure and psychometric properties of the tool are consistent across different age groups. (4) Conclusions: HSPS-10 is a simple and quick tool for group screenings as well as the individual assessment of school students and adults aged 12–25. The developed standardized procedure allows for the early recognition and identification of changes in the SPS over the course of life.

## 1. Introduction

The evaluation of the intensity of Sensory Processing Sensitivity (SPS) and Environmental Sensitivity (ES) is becoming an increasingly useful part of potential analysis for school students as well as adults [1,2,3]. The results of the research indicate the importance of high sensitivity both for the occurrence of psychological difficulties [4,5] and the exceptional potential of highly sensitive people [6]. Compared to their peers HSPs particularly benefit from a supportive environment and from a positive school transition [7,8]. In the assessment of children and adolescents, it is especially important to provide the possibility of using non-complex tools. The Polish short version of the Highly Sensitive Person scale (HSPS-10) was developed along with the adaptation of the full version of the questionnaire in Poland. HSPS-10 was developed on the basis of a study conducted with a group of adults aged 19–25 years. It contains 10 questions and allows for the attainment of a general score and a sensitivity profile described by a three-factor structure [9]. However, children and adolescents are a group of particular interest from the perspective of the practical use of the HSP scale. There is a separate version of the scale–the Highly Sensitive Child scale (HSC)–which was designed for children and adolescents [10], making it difficult to fully compare the results at different stages of development. Even though the results of studies with the use of this scale in the group of adults are already being developed, it was interesting to test how the 10-question scale would apply to the studies of children and adolescents.

The authors of the present study decided to describe the psychometric properties and examine the practical usefulness of the 10-question version of the HSP scale for both primary and secondary school students. Such a solution is supported by practical considerations, as well as the possibility of ensuring a full comparison of results obtained over the course of life. The short version of the HSPS questionnaire was administered using a 27-item version of the scale. Data were classified into EFAs and verified using item response theory (IRT). IRT verification of items classified into three factors in the EFA identified the 10 items with the highest informative value. The structure of the 10-item version of the HSP scale was verified in the CFA. The 10-item version of the HSP scale demonstrates a good fit of the three-factor model to the data [9]. A verification of the psychometric properties of the short version of the scale in a group of students also provided the grounds for the development of standards allowing for differential developmental diagnosis, and this is a gap worth filling among the various versions of the scale used in Poland and in the world.

Interest in researching SPS and ES enables the theoretical development of the issue, and also offers a high practical application value. The research conducted to date shows that the trait referred to as high sensitivity is associated with the risk of problems related to stress, as well as to psychological problems, when the person is brought up in an inappropriate, negative environment [11,12,13]. Among others, Liss et al. [14] showed that low and inadequate levels of parental care of highly sensitive children were significantly associated with their later depression. Similarly, childhood experiences of isolation, rejection or conflict may be associated with avoidant personality disorder (APD) in adult life [15]. In addition, this trait may be associated with special positive outcomes resulting from growing up and being raised and immersed in a positive and supportive environment [16,17,18]. High sensitivity is not a disorder, although it may resemble one. Therefore, it is important to have reliable and accurate tools to explore the significance of high sensitivity, properly identify it and provide adequate support to highly sensitive people.

## 2. Materials and Methods

### 2.1. Participants

In order to verify the psychometric properties of the short version of the HSP scale, studies were carried out on a group of primary school students and general and technical secondary school students of all class levels aged from 12 to 18 years. For the normalization of results, a comparative group of public and private university students from different faculties aged 19–25 was also included.

In addition to age, the stage of education and the type of secondary school, as well as the gender of the study participants, were used as criterion variables for the assessment of the level of SPS. Due to the diversity in the age of the students attending different types of schools (4-year general secondary school, 5-year technical secondary school), as well as the similar age of the students finishing primary school and those starting high school, the study participants were characterized across three age strata:(a)Up to 15.5 years—the transition from primary to secondary school education,(b)From 15.5 to 18 years—the stage before reaching the age of majority,(c)Persons of majority age (18 years and over).

The structure of the group of study participants is presented in Table 1. The group consisted of a total of 1384 people, with a slight predominance of males (55.7%). The sex ratio in each age range group was homogeneous (χ^2^(2) = 0.45; *p* = 0.800). Subgroups of students based on the type of school they attended differed significantly in their male-to-female ratio (χ^2^(3) = 40.94; *p* < 0.001).

The difference in sex structure between the subgroups of students distinguished by the different stages of education and the types of school attended resulted from the predominance of boys in technical and vocational schools (χ^2^(1) = 40.92; *p* < 0.001), which is typical for secondary education in Poland. The gender ratio among the groups of primary school and university students was homogeneous (χ^2^(1) = 0.001; *p* = 0.978).

### 2.2. Procedure

The Polish short version of the HSP scale was created as an adaptation of the Highly Sensitive Person scale–HSPS [19]. Developed by Elaine N. Aron [19] the tool consists of 27 questions. The scale has been translated into Polish (with the permission of the author of the scale and with the formal consent of American Psychological Association) using the back-translation procedure. At the first stage, the HSP scale was translated into Polish by two qualified psychologists with experience in psychometrics. The translation was then reviewed and translated back into English. The final version was translated again into Polish and then edited by a team of psychologists fluent in English so that the content of the test items was fully consistent with the Polish cultural context. The study participants answered questions using a 7-point Likert scale [9]. The research was conducted in groups; the pen-and-paper method was used among university students while primary and secondary school students answered the questionnaire online (Appendix A). An online survey was performed on a website accessible from school computer labs, which allowed for the submission of students’ answers to the database with an encrypted protocol, after an individual, randomly generated, anonymous password had been entered. The equivalence of paper and online responses was assumed, which was supported by many other studies [20,21,22,23].

University students filled out the questionnaires on their own, with instruction from the researchers. Among the school students, the survey was conducted under the supervision of school counsellors/psychologists or students in their final years of Pedagogical Sciences or Psychology, who were neither teachers nor did they have any other influence on the respondents. Minors were examined with the consent of the institutions in which the anonymous survey was conducted. Study participants were informed about the possibility of opting out of the study at any time, without needing to give a reason. The participants did not receive any reward. The study was conducted according to the guidelines of the Declaration of Helsinki, and approved by the Institutional Review Board of the University of Economics and Innovation in Lublin (No. 4/05/2020, 13 May 2020).

The analysis of the results was based on the validation of the fit of the model developed based on the available data. For the short version of the Highly Sensitive Person scale, the internal consistency of the scales was also checked. In addition, a set of standardized sten scores (sten in short) has been developed to ensure the comparability of individual results. The confirmatory factor analysis (CFA) of the scale structure was performed using a model that allows for the grouping of the answers to the 10 questions of the questionnaire into 3 factors, with a second-order factor being the general result of the short version of the HSP scale [9] In order to assess the internal consistency of the factors, the Cronbach’s α coefficient was used [24,25]. The thresholds of the sten scores were determined through the use of a linear transformation of the standardized results [26,27]. The divergence of the groups for which the score thresholds were required for the conversion into sten scores was established, and they were assessed using the Mann–Whitney U and Kruskal–Wallis H robust tests, independent of the shape of the distribution.

## 3. Results

### 3.1. Factor Structure of the HSPS-10 Questionnaire

The HSPS-10 Questionnaire was developed on the basis of the results of surveys conducted among university students aged 18–25 [9]. The possibility of using a shortened version (HSPS-10) of the Highly Sensitive Person scale with a group of people aged 12 to 25 years, who were students of primary and secondary schools and universities, was verified separately in each of the distinguished age layers. The results of the confirmatory factor analysis are presented in Table 2. The scale structure of the shortened HSP questionnaire is concordant with the data from the surveys carried out among primary school pupils from the age of 12, secondary school pupils and young adults up to the age of 25.

RMSEA indicators in all groups are similar and indicate a good fit in the whole group, as well as in the adult group, and an average (mediocre, according to one of the older terminologies) fit in the group up to 18 years of age [28]. Based on the NFI criterion and the single-sided PCLOSE test and AIC information criterion, it may be concluded that the highest degree of agreement of the scale structure with the data occurs in the sample analysed as a whole, without division into age groups and within the group of adults over 18 years of age. In turn, the CMIN/DF ratio indicates the best model match in relative terms within the group of students aged 15.5–18 years. The results, therefore, confirm that the psychometric value of the 10-item version of the HSPS scale is fully satisfactory across the age spectrum of the subjects from 12 to 25 years.

### 3.2. Reliability of the HSPS-10 Questionnaire

The reliability of the HSP questionnaire was assessed for the whole group as well as for the age subgroups. Coefficients of internal consistency for the whole group and age subgroups are presented in Table 3. The reliability of the overall HSP score ranges from 0.72 for students under 15.5 years old to 0.79 for university students, which may be considered a good score.

The internal compatibility of the EOE scale is at a similar level to the reliability of the overall score, for the EOE scale the values of the Cronbach’s α coefficient are between 0.757 for the group of students under 15.5 years and 0.81 for the group of adults. The internal consistency of the AES scale is lower and ranges from 0.57 for the group of students up to 15.5 years old to 0.67 for the group of secondary school and university students. The highest internal compatibility was determined for the LST scale, for which the Cronbach’s α coefficient is within the range of from 0.86 to 0.88. The internal coherence of the tool may be considered satisfying [25].

The results of the confirmatory factor analysis (CFA) and the reliability assessment indicate that the short version of the HSP questionnaire can be used for scientific research purposes as well as for routine screening and individual diagnosis in a group of people aged from 12 to 25 years. However, caution should be exercised when interpreting the AES scores, especially for pupils under 15.

### 3.3. Results for School Students of Different Ages and Sexes

The average scores of the short version of the HSP questionnaire that were determined for the study participants are presented in Table 4. In order to evaluate the degree of differentiation, a comparison of the HSP scores across age and gender groups and education profiles was carried out. Both sexes and ages in the groups were distinguished based on the stage of education and school type.

The distribution of the scores of the HSPS-10 scales differed from the normal distribution for the entire group of study participants and also in the subgroups distinguished based on the age and sex. A comparison between people from the distinguished age groups shows significant differences, both in terms of the overall result of the questionnaire and in the EOE and AES scales, while the results of the LST did not differ significantly between the three age ranges: HHSPS(2) = 30.17, *p* <0.001, HEOE(2) = 11.30, *p* = 0.004, HAES(2) = 64.84, *p* < 0.001; HLST(2) = 0.03; *p* = 0.986.

A comparison of the genders within the entire examined group and the three age groups is shown in Table 5. A comparison of girls/women with boys/men, regardless of age, indicates significant differences in the overall EOE score, but no differences were found in the AES and LST scales. Girls up to the age of 15.5 scored higher than boys in all of the HSPS-10 scales, as well as the general score. The differences between the other age groups are not significant.

The differences between the groups of the examined individuals distinguished by age range and sex, although not numerous, prove the advisability of converting the raw scores of the HSPS-10 scales into normalized results, thereby allowing for a comparison of values between gender and age groups. The need for the development of sten standards is also justified by the skewness of scale distributions. Separate norms were prepared for three age groups for the general scale, EOE and AES scales. Separate norms for girls and boys were also prepared for students aged 12–15.5. A set of sten standards for groups distinguished on the basis of the above-mentioned differences is presented in Table 6 and Table 7.

High sten scores of the HSPS-10 Questionnaire (sten scores between 7 and 10) indicate a heightened awareness of sensory stimulation, deeper cognitive processing of environmental stimuli, greater emotional and physiological reactivity and behavioural inhibition. Individuals with high trait scores show stronger activation of the autonomic nervous system in stressful situations, more intense positive and negative emotional reactions and feelings towards others, strong perception of subtle differences, knowledge of long-term consequences, and are characterised by a low threshold of sensitivity to external and internal stimuli and low tolerance to high levels of sensory stimuli [29,30].

A high score in the Ease of Excitation (EOE) scale indicates a tendency to feel overwhelmed in crowded places, difficult situations, or when having a lot to do in a short period of time. A Low Sensory Threshold (LST) is associated with a rapid response to a negative feeling caused by sensory stimuli such as loud sounds, bright lights, or touching. A high score on the Aesthetic Sensitivity (AES) scale indicates an awareness of subtleties in the environment, such as details, pleasant smells and tastes.

Average sten scores of the HSPS-10 Questionnaire (sten scores between 5 and 6) indicate a moderate awareness of sensory stimulation, cognitive processing of environmental stimuli and average emotional and physiological reactivity and behavioural inhibition. Individuals with average trait scores show relative resilience in stressful situations, moderate positive and negative emotional reactions and feelings towards others, average perception of subtle differences or knowledge of long-term consequences. They have a moderate threshold of sensitivity to external and internal stimuli and average tolerance to high levels of sensory stimuli [29,30].

An average score in the Ease of Excitation (EOE) scale indicates a moderate tendency to feel overwhelmed in crowded places, difficult situations, or when having a lot to do in a short period of time. An average score in Low Sensory Threshold (LST) is associated with a moderate response to a negative feeling caused by sensory stimuli such as loud sounds, bright lights, or touching. An average score on the Aesthetic Sensitivity (AES)scale indicates a moderate awareness of subtleties in the environment.

## 4. Discussion

The Highly Sensitive Person Scale (Aron and Aron, 1997) was developed for the evaluation of the SPS. Some of the researchers in the reports cited above used modified versions of the HSPS scale [31,32,33,34].

The research conducted to date has not clearly resolved the question of whether the characteristics widely known as high sensitivity are a homogeneous construct or whether they consist of several factors, while meta-analyses indicate that the three-factor model is the closest match that may be confirmed statistically and factually justified [35,36]. Research carried out to verify the psychometric properties of HSPS as well as its various aspects in a group of Polish young adults led to the construction of a proposal for a shortened version of the scale [9]. The best solution, from the perspective of psychometric analyses and the substantive basis, turned out to be the variant with the number of test items reduced to 10, and this scale was used in further research.

The results of the analyses that were carried out with all the studied age groups are similar and indicate a good fit to a three-factor solution for the whole group, as well as for the group of adults alone and an average mediocre fit for the group under 18 years of age, confirming the usefulness of the short version of the HSPS scale. To date, the HSC scale has been used with good results in studies on both children and adolescents [10,37,38], and also in the case where parents report answers concerning their child [39,40]. The HSC scale also indicates a three-factor solution, and the distinguishing factors include: Ease of Excitation, Aesthetic Sensitivity and a Low Sensory Sensitivity Threshold. This is the most reported solution supported by psychometric analyses and is consistent with the conceptual assumptions in both children’s and adult studies [10,33,35,41]. The abbreviated version was also used in research by other authors [33,34]. An analysis of gender differences revealed significant differences between boys/men and girls/women, but not for all factors and not for every age range. It revealed differences that are partially consistent with the results of other studies [19,41,42].

In the context of the research conducted, it is important to pay attention to the statistical methods of the analyses conducted to date, which explored the structure of the scale. Some of the studies cited above are based on a factor analysis performed using the Principal Component Method (PCA); sometimes this is used in combination with a Confirmatory Factor Analysis (CFA) and conducted using the same statistical samples. Exploratory Factor Analysis (EFA) and confirmation analysis were also performed on the two halves of the sample [32,41,43].

Based on the obtained results, which revealed a small number of differences between groups of respondents distinguished by age group and sex, it may be concluded that it is advisable to convert the raw results of the HSPS-10 scales into normalized results, thereby allowing for a comparison to be made between the values for gender and age groups. The need for the development of sten standards is also justified by the skewness of the scale distributions.

It is important to note the limitations of the study and use of the scale itself. Although the assessment of sensitivity is of both theoretical and application importance, further research on the improvement in the scales is also needed. From the studies conducted to date, we know that environmental sensitivity is related to other temperament and personality traits. Further research in this area is needed and justified. Data collected, e.g., by Hellwig and Roth [44], suggest that SPS highly overlaps with established personality traits, and its relationship with emotion recognition ability can be fully explained by Neuroticism and Openness to Experience. Similar results are also reported by other authors [36,41,45]. The duration of the research may also be important. Certainly, further analyses of the scale should also consider its stability.

## 5. Conclusions

Highly sensitive people tend to have more substantial reactions even to weak stimuli and require more time to return to a balanced state after experiencing emotional excitement [36]. Specific psychophysical traits enable highly sensitive people to process stimuli more deeply at all physical, interpersonal, emotional and cognitive levels compared to other people, and these processes may lead to a significant overload. High sensitivity is not a problem in itself, it is not a dysfunction or disorder, but knowing its benefits and drawbacks requires a degree of awareness and monitoring in order to ensure proper conditions for development and suitable, task-based actions [12,46]. The HSPS-10 scale is a simple and rapid tool that allows for a group screening diagnosis, as well as an individual assessment of students and adults aged 12–25. The structure and psychometric properties of the tool are consistent across different age groups, and the developed standards allow for the early recognition and identification of changes in Sensory Processing Sensitivity over the course of life. Examination using the HSPS-10 could be the basis for planning work with a highly sensitive person, as well as indicating the need for a possible broadening of the positive diagnosis [9].

## Figures and Tables

**Table 1 ijerph-19-15775-t001:** Structure of the research group.

Group				Stage of Education									
				Primary School (ISCED 2)		Technical Secondary School/Level 2 Vocational School (ISCED 3)		General Secondary School (ISCED 3)		University (ISCED 6–7)		Total	
				N	%	N	%	N	%	N	%	N	%
Age	<15.5	Sex	M	148	55.2	56	72.7	87	50.3	0	0.0	291	56.2
			F	120	44.8	21	27.3	86	49.7	0	0.0	227	43.8
	15.5–18.0	Sex	M	0	0.0	51	77.3	72	44.2	0	0.0	123	53.7
			F	0	0.0	15	22.7	91	55.8	0	0.0	106	46.3
	>18.0	Sex	M	0	0.0	57	71.3	56	48.3	244	55.3	357	56.0
			F	0	0.0	23	28.8	60	51.7	197	44.7	280	44.0
	Total, by sex	Sex	M	148	55.2	164	73.5	215	47.6	244	55.3	771	55.7
			F	120	44.8	59	26.5	237	52.4	197	44.7	613	44.3
	Total			268	100	223	100	452	100	441	100	1384	100

**Table 2 ijerph-19-15775-t002:** The fitting of the scales of the shortened version of the Highly Sensitive Person scale HSPS-10 to the results of primary and secondary school and university students based on confirmatory factor analysis (CFA).

Data in the Model	CMIN/DF	NFI (TFI)	CFI	RMSEA	PCLOSE *	AIC
Whole group (1384 people)	4.49 ***	0.96	0.97	0.05	0.468	212.00
Age < 15.5	2.77 ***	0.93	0.96	0.06	0.152	155.42
Age 15.5–18.0	1.87 **	0.91	0.96	0.06	0.195	125.72
Age > 18.0	2.55 ***	0.96	0.98	0.05	0.507	148.26

* *p* < 0.05 ** *p* < 0.01 *** *p* < 0.001.

**Table 3 ijerph-19-15775-t003:** Reliability of the HSP scale–the Cronbach’s α coefficients of the scales of the short version of the HSPS-10.

Group	Scales of the HSPS-10 Questionnaire			
	HSPS	EOE	AES	LST
Total	0.76	0.79	0.65	0.87
Age < 15.5	0.72	0.76	0.57	0.86
Age 15.5–18.0	0.73	0.77	0.67	0.87
Age > 18.0	0.79	0.81	0.67	0.88

**Table 4 ijerph-19-15775-t004:** HSPS-10 scale scores in different groups among study participants.

Group					Stage of Education									
					Primary School (ISCED 2)		Technical Secondary School/Level 2 Vocational School (ISCED 3)		General Secondary School (ISCED 3)		University (ISCED 6-7)		Total	
					M *	s	M *	s	M *	s	M *	s	M *	s
Age	<15.5	Sex	M	HSPS	38.27	10.18	37.16	11.33	40.05	9.18			38.59	10.15
				EOE	19.30	6.99	19.79	7.36	21.20	7.22			19.96	7.15
				AES	11.26	4.33	10.75	3.95	11.68	4.37			11.29	4.27
				LST	7.70	3.98	6.63	3.42	7.17	3.56			7.34	3.77
			F	HSPS	42.43	11.45	35.62	9.84	44.86	9.28			42.72	10.79
				EOE	22.28	7.41	18.90	5.62	23.05	5.72			22.26	6.73
				AES	11.84	4.46	10.67	4.83	13.26	4.36			12.27	4.52
				LST	8.30	4.05	6.05	2.38	8.56	3.68			8.19	3.83
	15.5–18.0	Sex	M	HSPS			42.82	10.39	42.94	10.97			42.89	10.69
				EOE			21.63	7.02	22.04	6.90			21.87	6.93
				AES			13.29	5.10	13.69	4.24			13.53	4.60
				LST			7.90	4.44	7.21	3.95			7.50	4.16
			F	HSPS			41.20	11.53	46.23	10.76			45.52	10.96
				EOE			23.27	9.10	23.57	6.57			23.53	6.93
				AES			9.87	4.63	14.63	4.69			13.95	4.95
				LST			8.07	4.46	8.03	3.86			8.04	3.93
	>18.0	Sex	M	HSPS			40.67	11.53	43.95	9.16	44.48	10.64	43.79	10.63
				EOE			20.53	7.75	21.25	6.89	22.56	6.52	22.03	6.82
				AES			13.02	4.67	14.36	4.53	14.05	3.98	13.93	4.20
				LST			7.12	3.69	8.34	4.06	7.88	3.46	7.83	3.60
			F	HSPS			42.70	9.74	50.37	9.38	41.00	10.80	43.15	11.07
				EOE			22.70	5.95	25.30	6.90	21.10	6.71	22.13	6.89
				AES			13.00	3.80	15.40	4.61	12.91	3.98	13.45	4.22
				LST			7.00	4.07	9.67	4.11	6.98	3.39	7.56	3.76

* The range of possible results for the HSPS: 10–70, EOE: 5–35, AES: 3–21, LST: 2–14.

**Table 5 ijerph-19-15775-t005:** Comparison of the results of the short version of the HSPS-10 for both sexes in different age groups.

Group	N_K_, N_M_	HSPS-10 Questionnaire Scales *							
		HSPS		EOE		AES		LST	
		*Z* **	*p*	*Z* **	*p*	*Z ***	*p*	*Z ***	*p*
Total	613, 771	2.99	0.003	3.26	0.001	0.88	0.081	1.43	0.153
Age < 15.5	227, 291	4.19	<0.001	3.56	<0.001	2.36	0.018	2.49	0.013
Age 15.5–18.0	106, 123	1.72	0.086	1.84	0.065	0.76	0.447	1.16	0.245
Age > 18.0	280, 357	0.54	0.592	0.52	0.602	1.47	0.141	0.85	0.395

* Statistically significant results are marked bold. ** Standardized values of Z for the Mann–Whitney U test.

**Table 6 ijerph-19-15775-t006:** Sten standards for students aged 12–15.5.

Sten	Sex							
	Girls				Boys			
	HSPS	EOE	AES	LST	HSPS	EOE	AES	LST
1	10–21	5–8	-	-	10–15	-	-	-
2	22–26	9–10	3–5	-	16–21	5–6	3–4	-
3	27–30	11–15	6–7	2–3	22–28	8–12	5–6	2
4	31–37	16–17	8–9	4–5	29–33	13–15	7–8	3–4
5	38–42	18–22	10–11	6–7	34–38	16–19	9–10	5–6
6	43–47	23–25	12–14	8–9	39–43	20–23	11–13	7–8
7	48–53	26–28	15–16	10–12	44–48	24–26	14–15	9–11
8	54–58	29–31	17–18	13	49–51	27–30	16–17	12–13
9	59–61	32–34	19–20	-	52–56	31–33	18–19	-
10	62–70	35	21	14	57–70	34–35	20–21	14

**Table 7 ijerph-19-15775-t007:** Sten standards for people aged 15.5–25.

Sten	Age						
	15.5–18			18–25			15.5–25
	HSPS	EOE	AES	HSPS	EOE	AES	LST
1	10–18	5–7	3	10–20	5	3–4	-
2	20–25	8–10	4–6	21–26	6–10	5–7	-
3	27–33	11–14	7	27–32	11–14	8	2–3
4	34–38	15–18	8–10	33–38	15–18	9–11	4
5	39–43	19–22	11–13	39–43	19–21	12–13	5–7
6	44–49	23–26	14–16	44–48	22–25	14–15	8–9
7	50–54	27–28	17–18	49–54	26–28	16–17	10–11
8	55–59	29–32	19–20	55–58	29–31	18–20	12–13
9	60–61	33–34	-	59–63	32–34	-	-
10	62–70	35	21	64–70	35	21	14–14

## Data Availability

Baryła-Matejczuk, Monika (2022), “Short Polish version of the Highly Sensitive Person Scale (HSPS-10)”, Mendeley Data, V2, doi: 10.17632/7mj932wh75.2. Some part of the data is not publicly available now due to reasons since it belongs to an ongoing project.

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
