# Peer review of "HSPS-10—Short Version of the Highly Sensitive Person Scale for Students Aged 12–25 Years"

_ijerph, 2022, doi:10.3390/ijerph192315775_

Round 1

Reviewer 1 Report

Dear authors,

Thank you for this article, and the opportunity to review it. 

The article is about the validation of a short version of the HSPS in Polish for adolescents and young adults (HSC10 for 12-25 year olds).

Regarding the content. 

1. Why talk about "functional diagnosis" if sensory processing sensitivity is not a pathology but a characteristic considered as a temperamental trait (which can generate positive as well as negative effects according to the literature)? I think it would be more appropriate to delete this notion of diagnosis in order not to create confusion.

2. The literature review is rather relevant but in my opinion some important references are missing: either in relation to the study of HSPS in children (Gere et al., 2009; Liss et al., 2005, 2008; Meyer & Carver, 2000); or in relation to the questions about the difference between the concept of sensory processing sensitivity and the Big Five profiles (Hellwig & Roth, 2021)

3. Finally, it is not clear to me how the 10 items were selected. You mention the 27-item scale and the translation procedure. But you do not mention how you selected the 10 items, which you should provide so that they can be understood and compared with the overall scale.

4. You write "The results of the analyses that were carried out with all of the studied age groups are similar and indicate a good fit to a three-factor solution for the whole group, as well as for the group of adults alone and a mediocre fit for the group under 18 years of age. This sentence is rather ambiguous and I don't really know what to understand: if the results are similar, why is the indicator for the group under 18 years old poor if all the others are considered good? It would be necessary to comment on this result, which seems to me to be essential for a true understanding of what it means for the validation of the under-18 scale... Can we consider that this short version is usable for the under 18s? Should it be considered that this version cannot be categorised into 3 factors for under 18s? This seems important to me for those who would like to use it, both scientifically and clinically.

Regarding the form and writing. 

The article is quite well written and the English seems perfectly understandable to me. Two things however:

1. you regularly use the word "Sten", which seems to me to mean "standard ten scores". However, you sometimes use it by indicating for example "a set of standardized sten scores" (it seems to me that there is a redundancy in this kind of formulation: page 3 line 123-124) but I am not an English native speaker and I can therefore be wrong. So check it out.

2. you have indicated "HEOS" (page 6 line 192). I think that in this place you meant "HEOE" (to refer to the EOE component).

For all these reasons I recommend minor revisions before acceptation.

Best regards 

Author Response

Dear Reviewer,

Thank you for the opportunity to revise our paper on ‘HSPS-10 - short version of the Highly Sensitive Person Scale for students aged 12-25 years’. Your suggestions have been very helpful for improving the manuscript.

I have included Your comment after this letter and responded to indicating how we addressed Your suggestion and describing the changes we have made. The revisions have been approved by all authors. The changes are marked in the ‘track changes’ mode in the paper.

We hope the revised manuscript will better suit and we thank you for your continued interest in our research.

Kind regards,

Monika Baryła-Matejczuk

Response to Reviewer #1 Comments

Comment 1: Why talk about "functional diagnosis" if sensory processing sensitivity is not a pathology but a characteristic considered as a temperamental trait (which can generate positive as well as negative effects according to the literature)? I think it would be more appropriate to delete this notion of diagnosis in order not to create confusion.

Response 1: Thank you for bringing this confusing sentence to our attention. The word diagnosis in the functional context was supposed to have a positive dimension, indicating the assessment of the potential. However, I fully understand the possibility of taking it differently than intended. We changed the terms to better match the meaning of high sensitivity.

Comment 2: The literature review is rather relevant but in my opinion some important references are missing: either in relation to the study of HSPS in children (Gere et al., 2009; Liss et al., 2005, 2008; Meyer & Carver, 2000); or in relation to the questions about the difference between the concept of sensory processing sensitivity and the Big Five profiles (Hellwig & Roth, 2021)

Response 2:Thank you very much for this comment. Literature has been supplemented in the text. It was included both in the introduction and in the discussion section.

Comment 3: Finally, it is not clear to me how the 10 items were selected. You mention the 27-item scale and the translation procedure. But you do not mention how you selected the 10 items, which you should provide so that they can be understood and compared with the overall scale.

Response 3: Thank you for noting this problem. We added an explanation on preparation of 10-item HSP scale: 

“The short version of the HSPS questionnaire was administered using a 27-item version of the scale. Data were classified into EFAs and verified using item response theory (IRT). IRT verification of items classified into 3 factors in the EFA identified the 10 items with the highest informative value. The structure of the 10-item version of the HSP scale was verified in the CFA. The 10-item version of the HSP scale demonstrates a good fit of the 3-factor model to the data”.

Comment 4: You write "The results of the analyses that were carried out with all of the studied age groups are similar and indicate a good fit to a three-factor solution for the whole group, as well as for the group of adults alone and a mediocre fit for the group under 18 years of age. This sentence is rather ambiguous and I don't really know what to understand: if the results are similar, why is the indicator for the group under 18 years old poor if all the others are considered good? It would be necessary to comment on this result, which seems to me to be essential for a true understanding of what it means for the validation of the under-18 scale... Can we consider that this short version is usable for the under 18s? Should it be considered that this version cannot be categorised into 3 factors for under 18s? This seems important to me for those who would like to use it, both scientifically and clinically.

Response 4: Thank you for this remark. The HSPS usability is central for us. We explained that the term “mediocre” is one of terms used in older psychometric classifications, what may be somewhat misleading.

We also clearly expressed that “The results therefore confirm that the psychometric value of the 10-item version of the HSPS scale is fully satisfactory across the age spectrum of the subjects from 12 to 25 years”. Consequently, we replaced the ‘mediocre” term by “average” in Discussion, explaining that our study ”confirms the usefulness of the short version of the HSPS scale”.

It is also worth noting that the term 'mediocre' referred to only one of the used indicators of the fit of the factor model to the data, whose value deviates from the threshold of good results by only 0.01. 'Mediocre' is also only one possible expression of the level of fit of the model to the data, and the results of the other indicators are satisfactory. Nevertheless, we wanted to draw researchers attention to this slight deviation of the results from the model, in order to guide future searches for the best possible version of the questionnaire.

Regarding the form and writing. 

The article is quite well written and the English seems perfectly understandable to me. Two things however:

  1. you regularly use the word "Sten", which seems to me to mean "standard ten scores". However, you sometimes use it by indicating for example "a set of standardized sten scores" (it seems to me that there is a redundancy in this kind of formulation: page 3 line 123-124) but I am not an English native speaker and I can therefore be wrong. So check it out.

  2. you have indicated "HEOS" (page 6 line 192). I think that in this place you meant "HEOE" (to refer to the EOE component).

Thank you for your comments. The term has been clarified, the error has been corrected.

Reviewer 2 Report

Dear Author

This manuscript was well-written with elaborate results and discussion.

The research was conducted in groups, the pen-and-paper method was used among university students while primary and secondary school students answered the questionnaire online. Do you think this variation in study procedure will affect the study results?

What time you chose for the study? Will this affect the results of your study?

Was there any limitations of the study? Mention that.

Author Response

Dear Reviewer,

Thank you for the opportunity to revise our paper on ‘HSPS-10 - short version of the Highly Sensitive Person Scale for students aged 12-25 years’. Your suggestions have been very helpful for improving the manuscript.

I have included Your comment after this letter and responded to indicating how we addressed Your suggestion and describing the changes we have made. The revisions have been approved by all authors. The changes are marked in the ‘track changes’ mode in the paper.

We hope the revised manuscript will better suit and we thank you for your continued interest in our research.

Kind regards,

Monika Baryła-Matejczuk

Response to Reviewer #2 Comments

Comment 1: The research was conducted in groups, the pen-and-paper method was used among university students while primary and secondary school students answered the questionnaire online. Do you think this variation in study procedure will affect the study results?

Response 1: Thank you for focusing our attention on this problem. We added explanation on comparability between online an paper-pencil versions: “The equivalence of paper and online responses was assumed, what was supported by many other studies [18–23]”.

Comment 2: What time you chose for the study? Will this affect the results of your study? Was there any limitations of the study? Mention that.

The research was carried out over a period of about 1.5 years - first, the study of adults; the research itself lasted 16 months; we did not assume that the results obtained in the study were affected by the duration of the study; however, taking into account the dynamic social changes, the criterion was introduced to the limitations of the study. Thank you very much for drawing attention to this. A paragraph related to research limitations has also been added.